# Graph Convolutional Semi-Supervised Cross-Modal Hashing

## ABSTRACT

Cross-modal hashing encodes different modalities of multi-modal data into a low-dimensional Hamming space for fast cross-modal retrieval. Most existing cross-modal hashing methods heavily rely on label semantics to boost retrieval performance; however, semantics are expensive to collect in real applications. To mitigate the heavy reliance on semantics, this work proposes a new semi-supervised deep cross-modal hashing method, namely, Graph Convolutional Semi-Supervised Cross-Modal Hashing (GCSCH), which is trained with limited label supervision. The proposed GCSCH first generates pseudo-multi-labels of the unlabeled samples using the simple yet effective idea of consistency regularization and pseudo-labeling. GCSCH designs a fusion network that merges the two modalities and employs Graph Convolutional Network (GCN) to capture semantic information among ground-truth-labeled and pseudo-labeled multi-modal data. Using the idea of knowledge distillation, GCSCH employs a teacher-student learning scheme that can successfully transfer knowledge from the fusion module to the image and text hashing networks. Empirical studies on three multi-modal benchmark datasets demonstrate the superiority of the proposed GCSCH over state-of-the-art cross-modal hashing methods with limited label supervision.

## CCS CONCEPTS

• **Information systems → Multimedia and multimodal retrieval**.

## KEYWORDS

Semi-supervised hashing, Cross-modal retrieval, Graph convolutional network

## 1 INTRODUCTION

Recent years have witnessed a huge surge of multimedia data [21, 30], e.g., images, texts, audios, and videos on the web. The potential semantic correlation among multi-modal data can be exploited to achieve cross-modal retrieval. Cross-modal retrieval [21, 23], which aims to search for relevant instances from one modality using a query from another modality has drawn increasing attention. In general, existing cross-modal retrieval methods [23] first project multi-modal data into a common subspace, then measure semantic similarities, and finally perform retrieval in this common subspace. The common subspace is often real-valued, and thus similarity measurement and retrieval suffer from high computation costs with

rapid increase of data [21]. Hashing [30] has attracted considerable interest for large-scale retrieval due to its obvious superiority in terms of storage and computation. Hashing [5, 18, 19, 30] learns hash codes that well preserve similarity structure of original data.

Hashing has been successfully applied to large-scale cross-modal retrieval by harvesting its benefits. Cross-modal hashing [47] maps multi-modal data into a common Hamming space, where efficient retrieval is performed. The shallow cross-modal hashing methods [2, 5, 6, 10, 16, 29, 42, 47] typically extract the hand-crafted or deep features using pre-trained network and then learn hash codes based on the extracted features of multi-modal data. The deep cross-modal hashing [8, 9, 14, 17, 32, 35, 37, 39, 40] has been developed to jointly perform feature learning and latent hash code learning in an end-to-end manner, and has shown superior to shallow cross-modal hashing. Deep cross-modal hashing [9] directly takes raw multi-modal data as inputs, e.g., raw image, bag-of-word text, and transforms them into hash codes using DNNs while incorporating semantic supervision. Typically, a similarity matrix is constructed based on whether two samples share common labels to indicate pairwise semantics. However, label semantics are expensive to obtain in real-world tasks, limiting the widespread application of deep supervised cross-modal hashing on cross-modal retrieval.

Semi-supervised cross-modal hashing [25, 33, 44, 45] mitigates the heavy reliance on labels by considering semantics of labeled multi-modal data and structure information of unlabeled multi-modal data. The hash functions are learned through the joint optimization of supervised losses on a small amount of labeled data and unsupervised losses on a vast amount of unlabeled data. However, semantics among unlabeled multi-modal data have not been exploited effectively, and the learned hash codes are not highly discriminative especially when labeled data is limited. Therefore, there remains a research gap in developing deep semi-supervised cross-modal hashing that is expected to yield improved retrieval performance.

To mitigate heavy reliance on labels and harness a vast mount unlabeled multi-modal data effectively, this paper proposes a new deep semi-supervised cross-modal hashing method, i.e., Graph Convolutional Semi-Supervised Cross-Modal Hashing (GCSCH) for cross-modal retrieval. The proposed GCSCH first predicts pseudo-multi-labels of unlabeled multi-modal data using consistency regularization, and further leverages the superior capability of Graph Convolutional Network (GCN) to effectively exploit semantic structure of the whole multi-modal data that can effectively supervise multi-modal hashing network training and hash code learning. The proposed GCSCH comprises three components including image/text network, consistency regularized pseudo-labeling module, and GCN fusion module, as shown in Figure 1. The main contributions of this work are as follows:

- We propose Graph Convolutional Semi-supervised Cross-modal Hashing (GCSCH) that is trained with limited label supervision for cross-modal retrieval. The proposed GCSCH

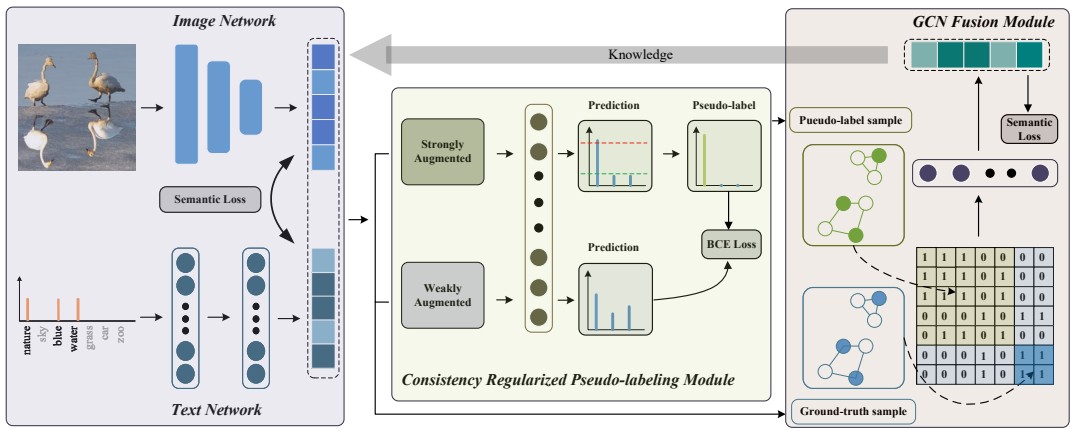

Figure 1: The illustration of the proposed Graph Convolutional Semi-Supervised Cross-Modal Hashing (GCSCH) for image-text cross-modal retrieval. The proposed GCSCH comprises three components including image/text network, consistency regularized pseudo-labeling module, and GCN fusion module. The image and text networks generate deep hash codes of image and text modalities respectively. The consistency regularized pseudo-labeling module generates pseudo-multi-labels for all the unlabeled multi-modal data with the popular idea of consistency regularization and pseudo-labeling. The GCN fusion module fuses image and text features and exploits semantic structure among ground-truth-labeled and pseudo-labeled multi-modal data by employing GCN. It further transfers knowledge of fused feature to guide training of image and text hashing networks via a teacher-student learning scheme.

can generate pseudo-multi-labels for all the unlabeled multi-modal data with the simple yet effective idea of consistency regularization and pseudo-labeling, such that the potential semantics of unlabeled data can be effectively exploited.

- The proposed GCSCH leverages GCN to effectively exploit semantic structure of ground-truth-labeled and pseudo-labeled multi-modal data, and effectively transfers the knowledge from fused feature to image and text networks using knowledge distillation.

- Extensive empirical results on three benchmark datasets demonstrate that the proposed GCSCH outperforms the state-of-the-arts on image-text retrieval with limited label supervision.

## 2 RELATED WORK

According to the amount of label semantic information used, cross-modal hashing [46] can be roughly divided into three categories: supervised cross-modal hashing [1, 26, 27, 31], unsupervised cross-modal hashing [22, 43], and semi-supervised cross-modal hashing [4, 24, 45].

Supervised cross-modal hashing methods indeed rely on the availability of labeled multi-modal data, which ensures that the learned hash codes are highly discriminative. This approach use the supervisory signals provided by the labels to guide the learning process, ensuring that the hash codes generated for different modalities are semantically consistent and can effectively capture the relevant information for discrimination tasks. Discrete Latent Factor Hashing (DLFH) [12] utilizes negative log-likelihood of cross-modal similarity, and optimizes in a discrete scheme that directly learns hash code without continuous relaxation. DLFH achieves

remarkable accuracy and trains much faster than some relaxation-based hashing methods. By leveraging the power of deep learning, deep cross-modal hashing integrates feature learning and hash code learning into a unified framework. For instance, Deep Cross-Modal Hashing (DCMH) [11] is among the first end-to-end learning frameworks with DNNs for cross-modal hashing. DCMH learns deep hash functions for each modality using DNNs and minimizes negative log-likelihood of cross-modal similarity. In contrast to supervised cross-modal hashing, unsupervised cross-modal hashing does not require label information that is often laborious to collect in real-world applications. Unsupervised deep cross-modal hashing (UDCMH) [34] combines deep learning and matrix factorization with binary latent factor models for multi-modal data retrieval. However, the performance of unsupervised cross-modal hashing is limited due to the lack of label supervision.

As labeled multimodal data is expensive to obtain in real-world scenarios, more efforts have been recently made towards semi-supervised cross-modal hashing, which considers both a small amount of labeled and a large amount of unlabeled multimodal data. For instance, Multi-view Graph Cross-modal Hashing (MGCH) [24] learns hash code in a semi-supervised manner using the outputs of multi-view graphs processed by a graph-reasoning module. Modality-specific and Cross-modal GCN (MCGCN) [33] employs two modality-specific channels and one cross-modality channel to learn modality-specific and shared representations for each modality respectively, and performs semantic information propagation from labeled data to unlabeled data via GCN. Semi-supervised Semi-paired Cross-modal Hashing (SSCH) [45] performs an alignment-free pseudo-labeling process that can strengthen semantic preservation to train effectively and efficiently. Semi-supervised Knowledge Distillation for Cross-modal Hashing (SKDCH) [25] utilizes teacher-student

optimization to propagate knowledge, and improves triplet ranking loss to better mitigate heterogeneity gap. The existing semi-supervised cross-modal hashing methods mainly consider semantics of labeled and structure of unlabeled multi-modal data, however semantics among unlabeled data have not been exploited effectively. In this work, we propose to exploit semantics among unlabeled multi-label data and employ such semantics to train multi-modal hashing networks.

## 3 APPROACH

This section presents the details of the proposed Graph Convolutional Semi-Supervised Cross-Modal Hashing (GCSCH), including problem setup, formulation, and out-of-sample extension.

### 3.1 Problem Setup

This work focuses on the image-text cross-modal retrieval. Assume that we are given a multi-modal dataset with $n$ image and text samples, where $n_l$ samples are labeled and $n_u$ samples are unlabeled, and we have $n = n_l + n_u$. Specifically, the labeled images and texts are denoted as $\mathbf{X}^l \in \mathbb{R}^{n_l \times d_x}$ and $\mathbf{Y}^l \in \mathbb{R}^{n_l \times d_y}$ respectively, and their labels are denoted as $\mathbf{L}^l \in \{0,1\}^{n_l \times c}$, where $c$ is the number of classes, $L_{ij}^l = 1$ if the $i$-th sample belongs to the $j$-th class and $L_{ij}^l = 0$ otherwise. In addition, the unlabeled images and texts are represented as $\mathbf{X}^u \in \mathbb{R}^{n_u \times d_x}$ and $\mathbf{Y}^u \in \mathbb{R}^{n_u \times d_y}$ respectively. The whole image and text modalities are represented as $\mathbf{X} = \left[\mathbf{X}^l, \mathbf{X}^u\right] \in \mathbb{R}^{n \times d_x}$ and $\mathbf{Y} = \left[\mathbf{Y}^l, \mathbf{Y}^u\right] \in \mathbb{R}^{n \times d_y}$ respectively. The goal of the proposed GCSCH is to learn the deep cross-modal hashing model and hash code $\mathbf{B} \in \{0,1\}^{n \times d}$ that supports efficient large-scale image-text cross-modal retrieval, where $d$ is hash code length.

### 3.2 Formulation

The proposed GCSCH is illustrated in Figure 1. As shown in Figure 1, GCSCH comprises three components including image/text network, consistency regularized pseudo-labeling module, and GCN fusion module.

#### 3.2.1 Image/Text Network.
For image modality, the proposed GCSCH employs Convolutional Neural Network (CNN) as the backbone to extract image features. Specifically, the proposed GCSCH obtains the high-level deep features of labeled and unlabeled images:

$$\mathbf{H}_X^l = f_X\left(\mathbf{X}^l, \Theta_X\right) \text{ and } \mathbf{H}_X^u = f_X\left(\mathbf{X}^u, \Theta_X\right) \quad (1)$$

where $\mathbf{H}_X^l \in \mathbb{R}^{n_l \times d}$ and $\mathbf{H}_X^u \in \mathbb{R}^{n_u \times d}$ denote labeled and unlabeled image features respectively, $f_X(\cdot)$ represents the image network, $\Theta_X$ denotes its network parameter. The feature of the whole image modality is represented by $\mathbf{H}_X = \left[\mathbf{H}_X^l; \mathbf{H}_X^u\right] \in \mathbb{R}^{n \times d}$.

For text modality, GCSCH uses Deep Neural Network as the backbone to extract text features. Specifically, the proposed GCSCH obtains the high-level deep features of labeled and unlabeled texts:

$$\mathbf{H}_Y^l = f_Y\left(\mathbf{Y}^l, \Theta_Y\right) \text{ and } \mathbf{H}_Y^u = f_Y\left(\mathbf{Y}^u, \Theta_Y\right) \quad (2)$$

where $\mathbf{H}_Y^l \in \mathbb{R}^{n_l \times d}$ and $\mathbf{H}_Y^u \in \mathbb{R}^{n_u \times d}$ denote labeled and unlabeled text features respectively, $f_Y(\cdot)$ represents the text network, $\Theta_Y$ denotes its network parameter. The feature of the whole text modality is represented by $\mathbf{H}_Y = \left[\mathbf{H}_Y^l; \mathbf{H}_Y^u\right] \in \mathbb{R}^{n \times d}$.

#### 3.2.2 Consistency Regularized Pseudo-labeling Module.
Leveraging unlabeled data to improve performance is key for semi-supervised learning (SSL). Instead of developing complex models, this work employs a simple yet effective module to accurately generate pseudo-multi-labels for unlabeled multi-modal data with consistency regularization [41]. In this work, following [36], we combine pseudo-labeling [13] and consistency regularization on two types of augmentations. We perform two types of feature augmentations: strong and weak, denoted by $\Lambda(\cdot)$ and $\lambda(\cdot)$ respectively. Specifically, we first concatenate the image and text features, and apply strong and weak augmentations on fused features using dropout with different parameters[20].

The loss in pseudo-labeling module includes two terms, i.e., a supervised loss $\mathcal{J}_{C_l}$ applied on labeled data, and an unsupervised loss $\mathcal{J}_{C_u}$ applied on unlabeled data. Specifically, the supervised loss is defined as the following loss on weakly augmented labeled fused feature:

$$\mathcal{J}_{C_l} = \frac{1}{n_l} \sum_{i=1}^{n_l} CE\left(f_C\left(\lambda([\mathbf{H}_X^l, \mathbf{H}_Y^l]_i), \Theta_C\right), \mathbf{L}_i^l\right) \quad (3)$$

where $f_C$ denotes a neural network for classification, and $\Theta_C$ is its network parameter, $\mathbf{L}_i^l$ is the multi-label vector of the $i$-th labeled sample, $CE$ denotes the widely-used standard cross-entropy loss.

For the unlabeled samples, we apply the idea of pseudo-labeling to first compute the multi-label distribution of strong augmented fused feature of the unlabeled samples $\mathbf{L}^u = f_C\left(\Lambda([\mathbf{H}_X^u, \mathbf{H}_Y^u]), \Theta_C\right)$. We then convert such multi-label distributions to their hard pseudo-multi-labels $\hat{\mathbf{L}}^u$, and the $j$-th label of the $i$-th unlabeled sample $\hat{L}_{ij}^u$ is set to 1 if it is larger than $\tau$, and set to 0 if it is smaller than $1-\tau$. We only retain pseudo-multi-labels of unlabeled samples whose label distributions all fall into such two regions defined by $\tau$ for training. With the popular idea of consistency regularization, we assume a good model should output similar predictions when fed different augmentations versions of one sample. Therefore, the unsupervised loss is defined as follows:

$$\mathcal{J}_{C_u} = \frac{1}{n_u} \sum_{i=1}^{n_u} I\left(\hat{\mathbf{L}}_i^u, \tau\right) CE\left(f_C\left(\lambda([\mathbf{H}_X^u, \mathbf{H}_Y^u]_i), \Theta_C\right), \hat{\mathbf{L}}_i^u\right) \quad (4)$$

where function $I(\mathbf{l}, \tau) = \prod_{i=1}^c \mathbb{1}\left(l_i \in [0, \tau] \cup [1-\tau, 1]\right)$ is used to select unlabeled samples with high-confidence prediction. We minimize the final loss that is defined as $\mathcal{J}_{C_u} + \mathcal{J}_{C_l}$ to train classifier in pseudo-labeling module, and further employ such learned classifier to generate pseudo-multi-labels of all the unlabeled samples $\mathbf{L}^u \in \{0,1\}^{n_u \times c}$ by setting $\tau$ to 0.5. To this end, we have labeled all the unlabeled samples, and the labels of all the samples are defined as $\mathbf{L} = [\mathbf{L}^l; \mathbf{L}^u]$.

The purpose of the proposed GCSCH is to preserve semantic similarity structure among image and text modalities. To achieve this goal, GCSCH proposes the following negative log-likelihood

loss to preserve semantic similarity among labeled and pseudo-labeled images and texts:

$$\min_{\Theta_X, \Theta_Y} \mathcal{J}_S = - \sum_{i,j=1}^{n} \left( S_{ij} \Omega_{ij}^{XY} - \log\left( 1 + e^{\Omega_{ij}^{XY}} \right) \right) \quad (5)$$

where $\Omega_{ij}^{XY} = \frac{1}{2} (H_X)_i (H_Y)_j^{\top}$, and $(H_X)_i$ and $(H_Y)_j$ are the $i$-th and $j$-th image and text features respectively. Specifically, based on $L$, we obtain semantic similarity matrix $S \in \{0, 1\}^{n \times n}$ to characterize similarity structure between all samples, and $S_{ij} = 1$ if the $i$-th and $j$-th samples share at least one common label, and $S_{ij} = 0$ otherwise.

In addition, to reduce information loss of quantizing continuous outputs of image and text networks, GCSCH minimizes the following quantization loss to enable continuous network outputs and hash codes to be close:

$$\min_{\Theta_X, \Theta_Y, B} \mathcal{J}_Q = \|H_X - B\|_F^2 + \|H_Y - B\|_F^2 \quad (6)$$

where $B$ denotes latent hash code shared by image and text modalities.

3.2.3 *GCN Fusion Module.* GCSCH proposes GCN fusion module to exploit complementary of multiple modalities and generate more discriminative hash code. Specifically, the proposed GCSCH concatenates the outputs of image and text networks, and further feeds the concatenation into fusion network to obtain high-quality fused feature. To further improve the quality of the fused features, the fusion module includes Graph Convolutional Network (GCN) to explore semantic information of both ground-truth and pseudo-labeled samples. With this adjacency matrix, GCN is then used to fully exploit semantics of labeled samples and structure of unlabeled samples. To alleviate over-smoothing problem that often occurs in GCN, the proposed GCSCH uses the following weighting scheme:

$$\tilde{S}_{ij} = \begin{cases} \left( S_{ij} / \sum_{j=1, i \neq j}^{n} S_{ij} \right) \times p, & i \neq j \\ 1 - p, & i = j \end{cases} \quad (7)$$

where $\tilde{S}$ is the weighted adjacency matrix, and $p$ is the weight assigned to the node itself and the other nodes. The proposed GCSCH first concatenates image and text features, and feeds it into a fusion network that consists of two-layer GCN:

$$H_F = f_F \left( \tilde{S}, [H_X, H_Y], \Theta_F \right) \quad (8)$$

where $f_F (\cdot)$ and $H_F \in \mathbb{R}^{n \times d}$ denote fusion network and fused feature respectively, $\Theta_F$ is its network parameter. The layer-wise propagation rule of GCN is defined as follows:

$$H^{(l+1)} = \sigma \left( \tilde{S}, H^{(l)}, \Theta^{(l)} \right) \quad (9)$$

where $H^{(l)}$ and $H^{(l+1)}$ denote the input and output of the $l$-th layer in fusion network respectively, $\Theta^{(l)}$ denotes the parameter of the $i$-th layer, and $\sigma (\cdot)$ denotes activation function. We define $H^{(1)} = [H_X, H_Y]$ and $H^{(3)} = H_F$. To further improve discrimination, the proposed GCSCH has the following negative log-likelihood loss on

---

**Algorithm 1** Graph Convolutional Semi-Supervised Cross-Modal Hashing (GCSCH)

---

**Input**: Labeled and unlabeled image $X^l$ and $X^u$; labeled and unlabeled text $Y^l$ and $Y^u$; label $L$; code length $d$; labeled and unlabeled size $m_l$ and $m_u$ in a mini-batch; parameters $\alpha, \beta, \gamma, \tau, p$.
**Output**: network parameters $\Theta_X, \Theta_Y, \Theta_F, \Theta_C$; hash code $B$.
1: Initialize $\Theta_X, \Theta_Y, \Theta_F, \Theta_C$;
2: **repeat**
3:     **for** $\max\{\lfloor \frac{n_l}{m_l} \rfloor, \lfloor \frac{n_u}{m_u} \rfloor\}$ iterations **do**
4:         Construct a mini-batch of $m_l$ labeled and $m_u$ unlabeled samples;
5:         Calculate image feature $H_X$, text feature $H_Y$, and fused feature $H_F$;
6:         Calculate the gradient of $\mathcal{J}_{C_u} + \mathcal{J}_{C_l}$ by chain rule, and update $\Theta_C$ by Adam algorithm;
7:         Apply pseudo-labels to all unlabeled samples;
8:         Calculate the gradient of $\mathcal{J}$ by chain rule, and update $\Theta_X, \Theta_Y, \Theta_F$, and $B$ by Adam algorithm;
9:     **end for**
10: **until** *Convergence*

---

labeled fused feature pairs:

$$\min_{\Theta_X, \Theta_Y, \Theta_F} \mathcal{J}_F = - \sum_{i,j=1}^{n_l} \left( S_{ij} \Omega_{ij}^{F} - \log\left( 1 + e^{\Omega_{ij}^{F}} \right) \right) \quad (10)$$

where $\Omega_{ij}^{F} = \frac{1}{2} (H_F)_i (H_F)_j^{\top}$, $(H_F)_i$ and $(H_F)_j$ denote the $i$-th and $j$-th fused features respectively. To improve the discrimination of outputs of image and text networks, under teacher-student learning framework [38], GCSCH regards fusion network and image/text network as teacher and student modules respectively, and uses fused feature generated by GCN to guide training of image and text networks. To achieve this, GCSCH minimizes the following simply but effective loss:

$$\min_{\Theta_X, \Theta_Y} \mathcal{J}_D = \|H_X - H_F\|_F^2 + \|H_Y - H_F\|_F^2 \quad (11)$$

With the above loss, the outputs of the image and text networks can be close to the fused features.

3.2.4 *Total Loss.* By combining the above four losses, i.e., $\mathcal{J}_S$, $\mathcal{J}_F$, $\mathcal{J}_D$ and $\mathcal{J}_Q$, we have the final objective function of the proposed GCSCH:

$$\min_{\Theta_X, \Theta_Y, \Theta_F, \Theta_C, B} \mathcal{J} = \mathcal{J}_S + \alpha \mathcal{J}_F + \beta \mathcal{J}_D + \gamma \mathcal{J}_Q \quad (12)$$

where the three parameters $\alpha$, $\beta$, and $\gamma$ are used to balance the importance of different losses.

The proposed GCSCH presents an iterative optimization scheme to find a feasible solution. For each iteration, we first optimize $\Theta_C$ to achieve better classification results and generate pseudo-labels, and then use pseudo-labels to optimize each of $\Theta_X, \Theta_Y, \Theta_F$ and $B$, while fixing the other variables. The proposed GCSCH can be trained in an end-to-end manner, and its overall training process is shown in Algorithm 1.

**Table 1: mAPs of all the cross-modal hashing methods on two cross-modal retrieval tasks with respect to 30% of labeled samples. The bold and underline indicate the best and the second best respectively.**

| Task | Method | Reference | MIRFLICKR-25K | | | | NUS-WIDE | | | | MS COCO | | | |
|------|--------|-----------|---------|---------|---------|---------|---------|---------|---------|---------|---------|---------|---------|---------|
| | | | 16 bits | 32 bits | 48 bits | 64 bits | 16 bits | 32 bits | 48 bits | 64 bits | 16 bits | 32 bits | 48 bits | 64 bits |
| I→T | DGCPN | AAAI 21 | 0.703 | 0.713 | 0.717 | 0.720 | 0.566 | 0.589 | 0.590 | 0.601 | 0.575 | 0.613 | 0.620 | 0.630 |
| | UCCH | TPAMI 23 | 0.734 | 0.741 | 0.739 | 0.739 | 0.590 | 0.610 | 0.615 | 0.618 | 0.562 | 0.569 | 0.569 | 0.590 |
| | LEMON | MM 20 | 0.651 | 0.670 | 0.668 | 0.682 | 0.460 | 0.491 | 0.512 | 0.507 | 0.492 | 0.438 | 0.522 | 0.527 |
| | EDMH | TKDE 22 | 0.651 | 0.657 | 0.655 | 0.646 | 0.460 | 0.477 | 0.474 | 0.461 | 0.502 | 0.497 | 0.468 | 0.427 |
| | HMAH | TMM 22 | 0.755 | 0.743 | 0.713 | 0.753 | 0.606 | 0.636 | 0.639 | 0.581 | 0.558 | 0.569 | 0.579 | 0.594 |
| | HCCH | TMM 23 | 0.719 | 0.730 | 0.737 | 0.736 | 0.625 | 0.638 | 0.643 | 0.649 | 0.560 | 0.606 | 0.621 | 0.634 |
| | MGCH | IS 22 | 0.689 | 0.705 | 0.694 | 0.729 | 0.525 | 0.514 | 0.557 | 0.595 | 0.615 | 0.562 | 0.549 | 0.607 |
| | SSCH | TCSVT 23 | 0.622 | 0.670 | 0.675 | 0.685 | 0.479 | 0.524 | 0.520 | 0.539 | 0.435 | 0.441 | 0.478 | 0.479 |
| | TS3H | TNNLS 23 | 0.717 | 0.741 | 0.751 | 0.742 | 0.613 | 0.642 | 0.650 | 0.671 | 0.618 | 0.624 | 0.648 | 0.690 |
| | GCSCH | Ours | **0.772** | **0.776** | **0.782** | **0.785** | **0.658** | **0.677** | **0.683** | **0.673** | **0.619** | **0.675** | **0.693** | **0.701** |
| T→I | DGCPN | AAAI 21 | 0.692 | 0.701 | 0.705 | 0.710 | 0.578 | 0.596 | 0.598 | 0.601 | 0.572 | 0.609 | 0.616 | 0.625 |
| | UCCH | TPAMI 23 | 0.722 | 0.726 | 0.722 | 0.725 | 0.600 | 0.616 | 0.623 | 0.626 | 0.553 | 0.560 | 0.559 | 0.586 |
| | LEMON | MM 20 | 0.666 | 0.695 | 0.687 | 0.708 | 0.472 | 0.508 | 0.538 | 0.517 | 0.487 | 0.475 | 0.528 | 0.535 |
| | EDMH | TKDE 22 | 0.668 | 0.677 | 0.679 | 0.667 | 0.475 | 0.487 | 0.490 | 0.477 | 0.501 | 0.494 | 0.464 | 0.427 |
| | HMAH | TMM 22 | 0.721 | 0.703 | 0.676 | 0.705 | 0.546 | 0.578 | 0.597 | 0.559 | 0.549 | 0.558 | 0.570 | 0.578 |
| | HCCH | TMM 23 | 0.721 | 0.740 | 0.748 | 0.742 | 0.631 | 0.632 | 0.639 | 0.649 | 0.556 | 0.588 | 0.620 | 0.647 |
| | MGCH | IS 22 | 0.675 | 0.695 | 0.684 | 0.719 | 0.541 | 0.515 | 0.553 | 0.607 | 0.601 | 0.553 | 0.524 | 0.586 |
| | SSCH | TCSVT 23 | 0.623 | 0.664 | 0.690 | 0.688 | 0.482 | 0.526 | 0.538 | 0.557 | 0.440 | 0.443 | 0.478 | 0.474 |
| | TS3H | TNNLS 23 | 0.727 | 0.753 | 0.764 | 0.748 | 0.622 | 0.653 | 0.665 | 0.674 | 0.614 | 0.618 | 0.645 | 0.687 |
| | GCSCH | Ours | **0.780** | **0.791** | **0.791** | **0.791** | **0.661** | **0.673** | **0.676** | **0.684** | **0.620** | **0.661** | **0.682** | **0.688** |

## 3.3 Out-of-Sample Extension

Once the proposed GCSCH is trained, the learned image and text networks can be used to generate hash code of a new query. Specifically, given an image query $\mathbf{x}_q$ or text query $\mathbf{y}_q$, its hash code can be generated as follows:

$$\mathbf{b}_q = \text{sign}\left(f_X\left(\mathbf{x}_q, \Theta_X\right)\right) \text{ or } \text{sign}\left(f_Y\left(\mathbf{y}_q, \Theta_Y\right)\right) \tag{13}$$

Once $\mathbf{b}_q$ is generated, it can be used for cross-modal retrieval by retrieving similar instances from database in another modality.

## 4 EXPERIMENTS

This section evaluates effectiveness of the proposed GCSCH by performing image-text cross-modal retrieval. The experiment is conducted on an Ubuntu Enterprise 64-bit Linux server equipped with an NVIDIA A6000 GPU. The proposed GCSCH is implemented using PyTorch.

## 4.1 Experimental Setup

*4.1.1 Datasets.* The experiment is conducted on three multi-label image benchmarks, i.e., MIRFLICKR-25K [7], NUS-WIDE [3], and MS COCO [15]. The details of three benchmark datasets are as follows:

*MIRFLICKR-25K*[1] [7] consists of 25,000 images collected from Flickr website. Each image is associated with several textual tags. Hence, each point is a image-text pair which is manually annotated with 24 unique labels. The text for each point is represented as a 1,386-dimensional bag-of-words (BoW) vector. The 20,015 points that have at least 1 label and 1 textual tag are selected for experiment. The 2,000 samples are randomly selected as a query set and the rest 18,015 samples are used as database. The 10,000 samples are randomly selected from database for training.

*NUS-WIDE*[2] [3] contains 260,648 web images, and some images are associated with textual tags. Each image is annotated with one or multiple labels from 81 concept labels. We select 195,834 image-text pairs that belong to the 21 most frequent concepts. The text for each point is represented as a 1,000-dimensional bag-of-words vector. The 2,100 samples are randomly selected as a query set and the rest 193,734 samples are used as database. The 10,500 samples are randomly selected from database for training.

*MS COCO*[3] [15] is constituted by two subsets of images: a training set with 82,783 training images and a validation set with 40,504 images. In our experiments, we merge the training images and validation images and remove those samples that have no text data. Finally, 122,218 image-text pairs are left for our experiments. The text for each point is represented by a 2,000-dimensional bag-of-words vector. The 5,000 samples are randomly selected as a query set and the rest samples are used as database. The 10,000 samples are randomly selected from the database for training.

*4.1.2 Baselines.* We compare the proposed GCSCH with nine state-of-the-art cross-modal hashing baselines for comparison, including three semi-supervised cross-modal hashing methods, i.e., SSCH [45], MGCH [24], TS3H [4], two unsupervised cross-modal hashing method, i.e., DGCPN [43], UCCH [22], and four supervised cross-modal hashing method, i.e., EDMH [1], LEMON [31], HMAH [27], HCCH [26]. All samples are used for training semi-supervised and unsupervised methods, while only labeled samples are used for training supervised methods.

---

[1]http://lear.inrialpes.fr/people/guillaumin/data.php

[2]http://lms.comp.nus.edu.sg/research/NUS-WIDE.htm
[3]http://mscoco.org/

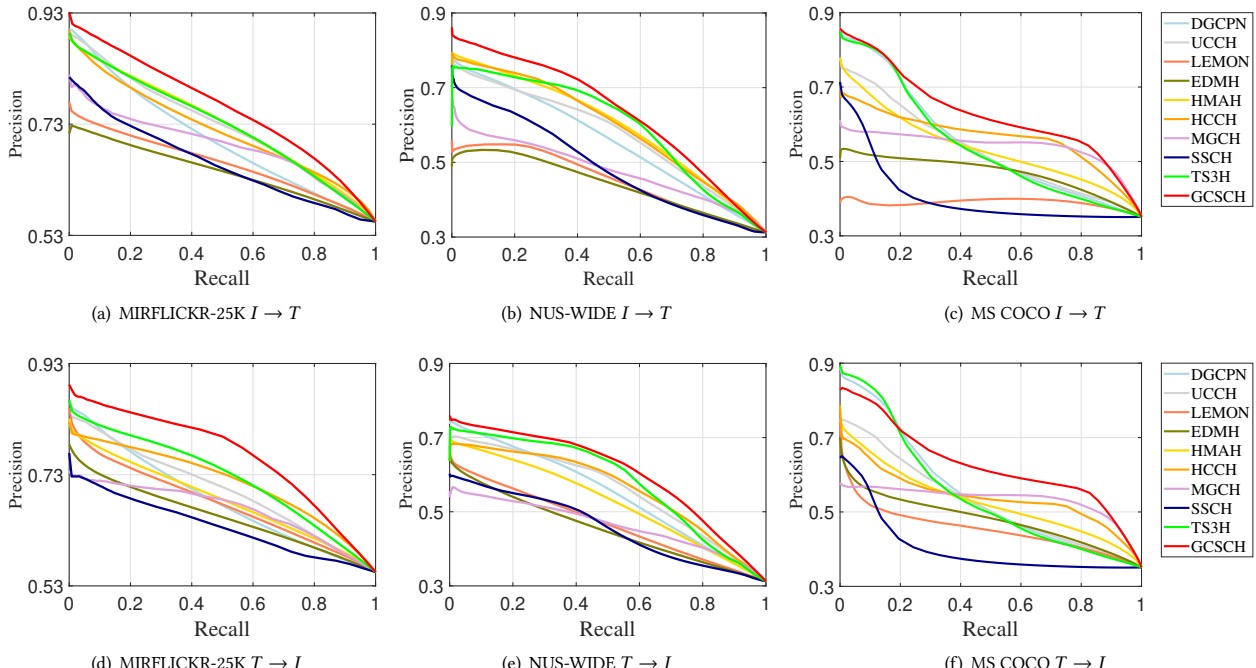

**Figure 2: Precision-recall curves of all the cross-modal hashing methods on two cross-modal retrieval tasks with respect to 30% of labeled samples.**

*4.1.3 Experiment Setting.* For the proposed GCSCH, the batch size is set to 128, where the numbers of labeled and unlabeled samples are set to 80 and 48 respectively, and the number of iterations is set to 100. The five trade-off parameters, i.e., $\alpha$, $\beta$, $\gamma$, $\tau$, $p$ are set to 0.1, 0.05, $10^{-9}$, 0.05, and 0.3 respectively. The Adam optimizer is used for optimizing. For image modality, we employ VGG model pre-trained on ImageNet as the backbone, where the classification layer is replaced by hash layer whose output dimension is set to $d$. For text modality, we employ text network with three fully-connected layers including a 1,024-dimensional hidden layer. The modality fusion network is a GCN network including two 1,024-dimensional graph convolutional layers and one 1,024-dimensional fully-connected layer. For all networks, ReLU is adopted as activation for hidden layers and tanh activation is used to approximate hash code. Weakly augment and strongly augment use different proportions of dropout operations.

*4.1.4 Evaluation Metrics.* Two cross-modal retrieval tasks are used for evaluation, i.e., I→T that retrieves relevant texts in database given any image query, T→I that retrieves relevant images in database given any text query. We consider the widely used metric, i.e., mean Average Precision (mAP) to evaluate retrieval performance, which is calculated using all the samples in databases.

## 4.2 Performance Evaluation

*4.2.1 Evaluation on Small Percentage of Labeled Samples.* This section compares the proposed method with state-of-the-art hashing baselines with a small percentage of labeled samples. We set the percentage of labeled samples in training set to 30%, and report

the mAPs of all the hashing methods with respect to different bits, i.e., 16, 32, 48, 64 bits in Table 1, where bold and underline indicate the best and second best in each case. The PR curves of all the methods with respect to 32 bits are shown in Figure 2. From Table 1 and Figure 2, we can clearly observe that (1) The proposed GCSCH has the highest mAPs among 24 cases, and outperforms the best baselines averagely by 5.19%, 3.62%, 3.65% on MIRFLICKR-25K, NUS-WIDE, MS COCO respectively. In addition, PR curves of GCSCH are generally above those of the baselines. (2) Among the semi-supervised baselines, TS3H outperforms the other two semi-supervised baselines. MGCH achieves similar performance, and SSCH underperforms in the most cases. (3) The unsupervised baselines are generally competitive to the supervised baselines in the setting of limited label information, among which HCCH performs best.

*4.2.2 Evaluation on Different Percentages of Labeled Samples.* This section compares the proposed method with the semi-supervised baselines with varying percentages of the labeled samples. We set code length to 32, and vary the percentages of labeled samples from the range of [10%, 90%]. The mAPs of all the semi-supervised methods with respect to different percentages are reported in Figure 3. As can be observed, the proposed GCSCH outperforms the semi-supervised baselines in most cases, indicating its superiority of learning from varying percentages of labels. As the percentage of labeled data increases, the mAPs of the proposed GCSCH improves stably, and consistently higher than the most baselines. Some baselines, e.g., MGCH show some performance fluctuations

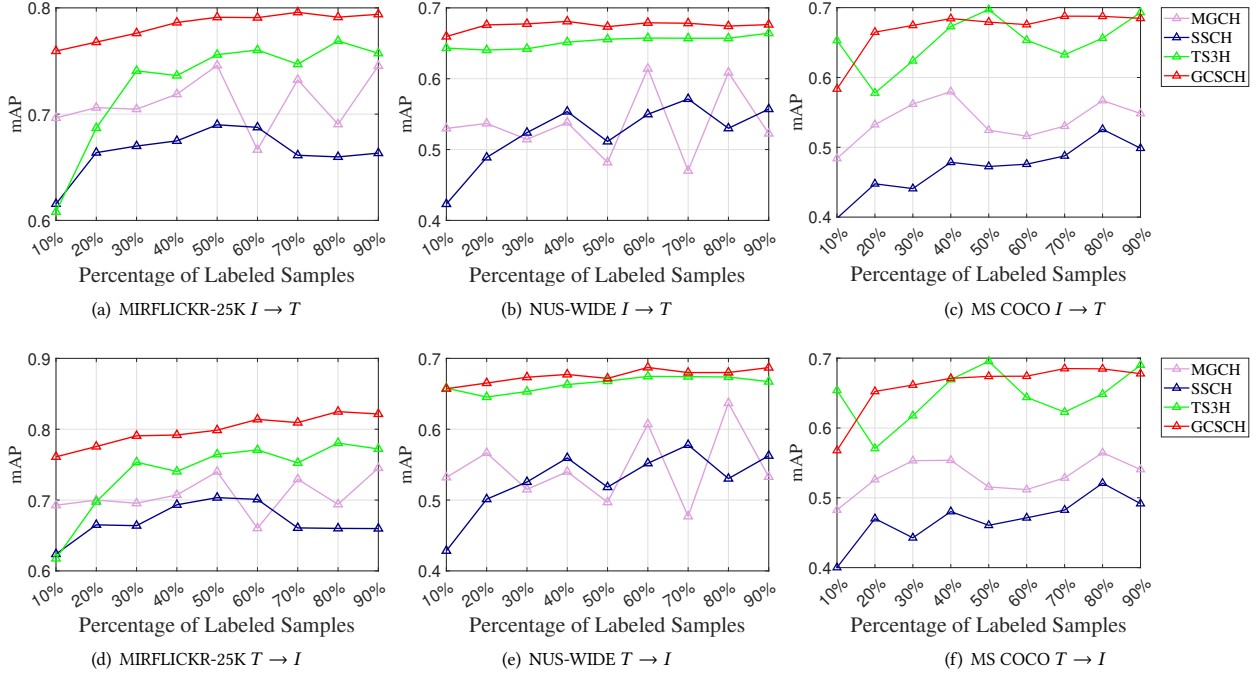

Figure 3: mAPs of the semi-supervised cross-modal hashing methods on two cross-modal retrieval tasks with respect to different percentages of labeled samples.

Table 2: Ablation study of the proposed GCSCH on NUS-WIDE. The bold indicates best.

| Task | Method | 16 bits | 32 bits | 48 bits | 64 bits |
|---|---|---|---|---|---|
| I→T | GCSCH-F | 0.6558 | 0.6591 | 0.6464 | 0.6409 |
| | GCSCH-C | 0.5992 | 0.6122 | 0.6206 | 0.6194 |
| | GCSCH | **0.6577** | **0.6774** | **0.6825** | **0.6730** |
| T→I | GCSCH-F | 0.6484 | 0.6538 | 0.6391 | 0.6484 |
| | GCSCH-C | 0.6202 | 0.6363 | 0.6436 | 0.6391 |
| | GCSCH | **0.6607** | **0.6734** | **0.6764** | **0.6836** |

with the change of the percentage of labeled data. The above empirical results show that the proposed GCSCH can effectively handle partially labeled multi-modal data.

## 4.3 Further Analysis

*4.3.1 Ablation Study.* We conduct ablation study of the proposed method by comparing it with its two variants. GCSCH-F is a variant of GCSCH that removes GCN Fusion Module. GCSCH-C is a variant of GCSCH that removes Consistency Regularized Pseudo-labeling Module. We adopt NUS-WIDE for experiment, set the percentage of labeled samples to 30%, and report the mAPs of all the methods on the two cross-modal retrieval tasks in Table 2. As can be observed, the proposed GCSCH clearly outperforms its two variants among all the cases. Specifically, GCSCH improves GCSCH-F averagely by 1.09%, 2.89%, 5.71%, 5.22% with respect to 16, 32, 48, 64 bits respectively. It indicates that fusion module obviously improves quality of

hash code by effectively guiding the training of hashing networks. GCSCH improves GCSCH-C averagely by 8.12%, 8.19%, 7.49%, 7.79% with respect to 16, 32, 48, 64 bits respectively. It demonstrates that pseudo-labeling module can generate accurate pseudo-labels that provide strong semantic supervision and improve discrimination of hash code. The above empirical results clearly demonstrate the effectiveness of pseudo-labeling and fusion modules.

*4.3.2 Parameter Sensitive Analysis.* We empirically analyze the sensitivity of the five parameters, i.e., $\alpha$, $\beta$, $\gamma$, $\tau$, $p$ in the proposed GCSCH. Specifically, $\alpha$, $\beta$, and $\gamma$ determine the relative importance of each loss, $\tau$ determines the threshold for generating pseudo-labels, and $p$ determines the weights of a node itself and the other nodes in graph construction. We adopt NUS-WIDE for experiment, and set the percentage of labeled samples and code length to 30% and 32 respectively. The mAPs of the proposed GCSCH with respect to different parameters are shown in Figure 4. From this figure, we see that mAPs are generally relatively stable to the change of $\alpha$, $\gamma$, $\tau$ and $p$. The parameter, i.e., $\beta$ has relatively high impact on the performance of GCSCH. As $\beta$ increases, the mAPs first improve and then drop, and highest mAP is obtained when $\beta$ is set to 0.05. It verifies effectiveness and stable of the proposed fusion network.

*4.3.3 Visualization.* We visualize the learned hash code to qualitatively verify the proposed method, and compare the proposed method with the hashing baselines. We adopt NUS-WIDE for experiment, and 8,000 samples that are annotated with only one label are randomly selected and code length is set to 32. The hash

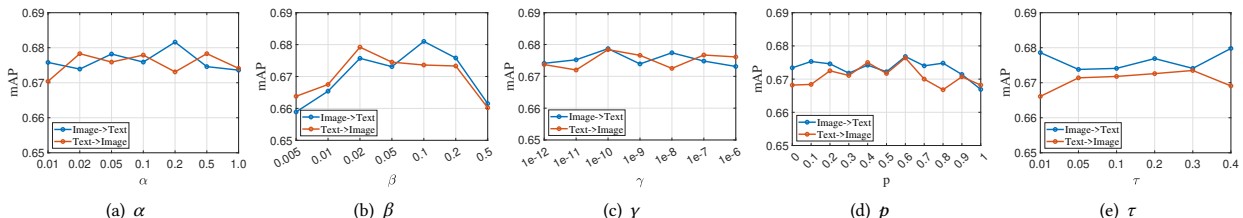

Figure 4: Parameter sensitivity analysis of the proposed GCSCH on NUS-WIDE.

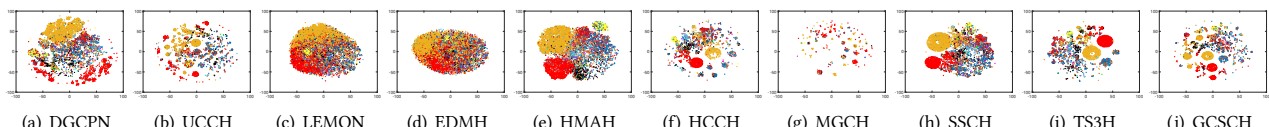

Figure 5: The t-SNE visualization of NUS-WIDE using all the hashing methods.

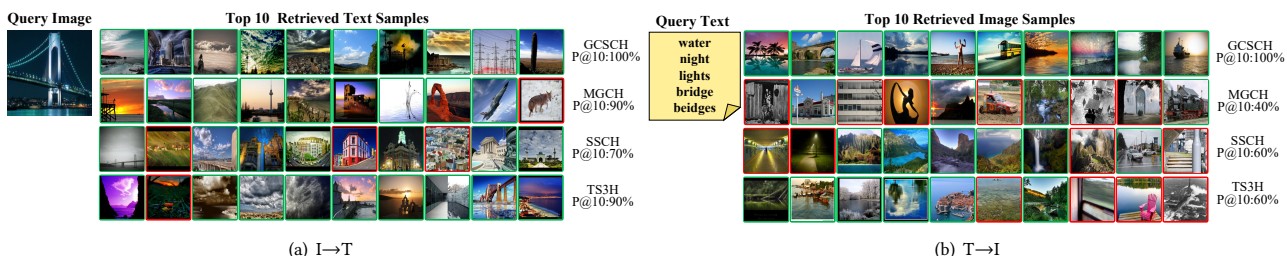

Figure 6: Top-10 retrieved results of the semi-supervised cross-modal hashing methods on a randomly selected image and text query pair from NUS-WIDE.

codes learned by the ten hashing methods are visualized into a 2-dimensional space with t-SNE [28], as illustrated in Figure 5. From Figure 5, we see that visualization of the proposed GCSCH is better than the baselines, which is generally consistent with previous quantitative empirical results.

4.3.4 *Case Study.* We present a case study of image-text cross-modal retrieval, and compare the proposed GCSCH and three semi-supervised baseline methods, i.e., MGCH, SSCH, TS3H. We adopt NUS-WIDE for experiment, and set code length and the percentage of labeled samples to 32 and 30% respectively. The top 10 retrieved results of a random image and text query pair on two cross-modal retrieval tasks are illustrated in Figure 6. Given an image query, the corresponding images of the retrieved texts are shown to enable retrieved results to be intuitive in I→T task. The retrieved sample is marked green if it shares at least one common label with the query, and is marked red otherwise. As can be seen from Figure 6, compared to the semi-supervised cross-modal hashing baselines, the proposed GCSCH obviously retrieves more similar samples on the two retrieval tasks. The above results qualitatively verify the effectiveness of the proposed method for image-text cross-modal retrieval.

## 5 CONCLUSION

This work studies semi-supervised cross-modal hashing with limited semantic supervision for cross-modal retrieval, and proposes Graph Convolutional Semi-Supervised Cross-Modal Hashing (GC-SCH) to mitigate heavy reliance on semantics. Compared to existing semi-supervised cross-modal hashing, this work can generate pseudo-multi-labels of unlabeled samples using the simple yet effective idea of consistency regularization and pseudo-labeling. In addition, this work fuses image and text modalities, employs GCN to capture semantic information among ground-truth-labeled and pseudo-labeled multi-modal data, and guides training of multi-modal hashing networks under teacher-student learning framework. Empirical studies on three benchmarks demonstrate the superiority of the proposed method over the state-of-the-arts in image-text retrieval with limited labels.

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
