# OpenReview forum: "Graph Convolutional Semi-Supervised Cross-Modal Hashing"
_acmmm.org/ACMMM/2024/Conference — MM2024 Poster_

### Official Review · Reviewer_vo6v · 2024-05-13

**Rating:** 3
**Confidence:** 2

**Summary:**

This paper proposes an effective cross-modal hashing method that combines elements from existing approaches to enhance cross-modal hashing performance across multiple datasets, achieving significant advantages when using a smaller proportion of labelled data.

**Strengths:**

The paper is readable, with a well-organized structure, and the experimental setup is generally complete. The innovation of the paper is moderate and incremental.

**Limitations:**

The main innovation and the module that contributes to performance improvement focuses on the Consistency Regularized Pseudo-labeling Module. This module enhances the quality of pseudo-labels through optimising consistency between strong and weak data augmentation expressions, however, which is not a novel concept. Ablation study results suggest that integrating more advanced technologies into the pseudo/real label graph model fusion does not significantly enhance results compared to the Consistency Regularized Pseudo-labeling Module, serving more as a complement to it.

Some issues are:

1. Could you provide more specific definitions for strong/weak augmentation?
2. The authors claim that the GCN Fusion Module makes hash encoding more distinctive. In the experimental section, Figure 5 visualises the hash code distributions of different methods. A more interesting result could be the visualisation of GCSCH compared to GCSCH-F (perhaps also GCSCH-C).
3. The weight \beta of the L_D loss in the knowledge distillation module significantly impacts the results, which is understandable. However, a more detailed analysis is needed to understand what this implies and possible reasons behind it.

**Suitability:**

2

---

### Official Review · Reviewer_6ohf · 2024-05-22

**Rating:** 6
**Confidence:** 3

**Summary:**

This paper proposes a novel semi-supervised cross-modal hashing method, i.e., Graph Convolutional Semi-Supervised Cross-Modal Hashing (GCSCH). The proposed method can be trained with limited label supervision for cross-modal retrieval. Specifically, a simple yet effective idea of consistency regularization and pseudo-labeling is introduced to generate pseudo-multi-labels for all the unlabeled multi-modal data; and GCN is adopted to effectively exploit semantic structure of ground-truth-labeled and pseudo-labeled multi-modal data. Experimental results demonstrate that the proposed method can outperform SOTA methods.

**Strengths:**

1. This paper is well written and easy to follow.

2. The proposed method is reasonable and sound. The idea of using consistency regularization and pseudo-labeling is interesting in semi-supervised cross-modal hashing.

3. Experimental results demonstrate that the proposed method can outperform SOTA methods on image-text retrieval with limited label supervision.

**Limitations:**

1. It is mentioned that there is a strong augmentation and a weak augmentation. However, related description in the paper is only to augment the fused features with different dropout parameters. It is suggested to provide a more detailed description of this feature augmentation operation.

2. The ablation experiment only involves removing the fusion module and the pseudo-label module separately, which is not very sufficient to show the effect of the proposed method. It is recommended to add extra experimental results about removing both modules.

3. It is suggested to check the paper to correct minor typos.

**Suitability:**

3

---

### Official Review · Reviewer_Dywf · 2024-05-24

**Rating:** 2
**Confidence:** 4

**Summary:**

This paper proposes GCSCH to solve the cross-modal hashing retrieval problem under limited supervision. The proposed method employs three main approaches, including pseudo-labeling, GCN and knowledge distillation.

**Strengths:**

1. The paper is easy to follow.
2. Experimental results demonstrate the effectiveness of the proposed method.

**Limitations:**

1. The novelty of this paper is incremental, as the proposed framework just integrates some well-developed approaches for performance improvement.

2. The construction of S is based on the available labels. Then, S will be a very sparse matrix, as very limited samples have labels. In this case, I am confused if the loss functions Eq. (5) and Eq. (10) can be useful with a very sparse adjacency matrix. Or does the construction of S use the pseudo-labels of unlabeled samples?

3. The authors only provide two abstract functions for feature augmentations in Line 304. How to implement these two functions in the experiments? No details are provided.

4. In Fig. 3, in some cases, why does using more labels lead to a performance drop?

5. From Fig. 5, it seems that TS3H is better than the proposed GCSCH.

6. If the parameter \gamma is set to 10^-9, does it mean the related part can be ignored with regard to the overall loss?

**Suitability:**

3

---

### Official Review · Reviewer_7Krx · 2024-05-26

**Rating:** 3
**Confidence:** 3

**Summary:**

This paper proposes a novel semi-supervised deep cross-modal hashing method to capture semantic information among ground-truth-labeled and pseudo-labeled multi-modal data based on graph convolutional network. Experimental results show the effectiveness of the proposed approach. The following concerns should be addressed.

**Strengths:**

1.The method has achieved significant improvements in experiments.
2.This paper is well organized and written.

**Limitations:**

Some concerns and suggestions to enhance this work are as follows:
1. The motivation in the abstract is unclear. I strongly suggest that the description need to be enhanced clarity and succinctness
2. I recommend the author to put the notations into figure 1 as well and further improve the clarity and readability of them.
3.The introduction in manuscript lacks effective organization and coherence. I encourage the authors to carefully revise the manuscript with a focus on improving its overall organization and writing structure. Avoid overly general descriptions, such as “ [2, 5, 6, 10, 16, 29, 42, 47]” and “ [8, 9, 14, 17, 32, 35, 37, 39, 40]”.
4. I noticed several formatting errors such as the lack of punctuation in the formulas in Section 3.2.
5. It is recommended that the authors the language expression errors and enhance the manuscript's overall quality. Such as “Using the idea of knowledge distillation,” in the abstract.

**Suitability:**

2

---

### Meta-Review · Area_Chair_tzon · 2024-07-01

**Recommendation:** Accept (Poster)
**Confidence:** 4

**Metareview:**

In this paper, the authors propose a semi-supervised method, Graph Convolutional Semi-Supervised Cross-Modal Hashing (GCSCH). This method uses pseudo-labeling, a fusion network, and a teacher-student learning scheme to capture semantic information and transfer knowledge, demonstrating superior performance on three benchmark datasets with limited label supervision. After the rebuttal phase, all the reviewers unanimously conclude acceptance. However, a few minor issues still need to be addressed before the camera-ready version. Please refer to the reviewers' limitations and final thoughts.